# Entrepreneurial intention of students: The role of digital tools and personal factors in entrepreneurship education

**Aivars Spilbergs[1], Inese Mavlutova[2], Kristaps Lesinskis[3]***

1 BA School of Business and Finance, Department of Economics and Finance, University of Latvia, Riga, Latvia, 2 BA School of Business and Finance, Department of Economics and Finance, University of Latvia, Riga, Latvia, 3 BA School of Business and Finance, Department of Management, University of Latvia, Riga, Latvia

* Kristaps.lesinskis@lu.lv

## Abstract

Education is changing due to digital transformation, which is especially significant for entrepreneurship since students' readiness to start their own businesses may be assessed by their proficiency with digital technologies. An experimental group educated with the AI-based digital tool KABADA, and a control group trained through traditional workshops in a quasi-experiment with 819 students from Southern and Central-Eastern Europe were compared to examine this relationship. The effects of gender, education, past entrepreneurial experience, self-evaluation, and motivation on entrepreneurial intention were tested using ordinal logistic regression. Compared to traditional workshops, the results demonstrate that digital tools significantly increase students' entrepreneurial intention; the best indicators were self-assessment and entrepreneurial desire. While education level had no discernible impact, gender and previous entrepreneurship experience were also important factors. The findings show that one of the most effective ways to encourage entrepreneurial purpose is to incorporate digital tools into entrepreneurship education. This study emphasizes the importance of students acquiring digital skills in order to increase their preparedness for entrepreneurship in technology-driven economies.

## Introduction

Since universities are crucial in forming the next generation of entrepreneurs, entrepreneurship education has emerged as a pillar of economic growth. As digital transformation gains momentum, new technologies are changing how students study, work together, and become ready for careers in entrepreneurship. Although interactive tools, digital platforms, and artificial intelligence provide up possibilities for more interesting and productive learning settings, little is known about how these technologies affect students' intentions to start their own businesses. Higher education

**Data availability statement:** Available in Argos Openaire platform. In working on this study, the authors adhere to the FAIR principles, all data are available on the Argos platform at this link: https://argos.openaire.eu/explore-plans/overview/public/034fd469-2c69-4208-bca5-c3151ba8a427.

**Funding:** This study is funded by the Recovery and Resilience Facility project "Internal and External Consolidation of the University of Latvia" (No.5.2.1.1.i.0/2/24/I/CFLA/007), Nr. LU-BAPA. -2024/1-0068. The funders had no role in study design, data collection and analysis, decision to publish, or preparation of the manuscript.

**Competing interests:** The authors have declared that no competing interests exist.

institutions and governments must comprehend how digital tools affect skills, motivation, and preparedness to launch a firm. With an emphasis on how digital tools, in conjunction with contextual and personal factors, influence students' intention to engage in entrepreneurial activity, this study explores the impact of digital transformation in entrepreneurship education.

The field of entrepreneurship education (EE) has undergone and will continue to undergo a digital transformation. New digital technology has made significant improvements in teaching conceivable. EE ensures the formation of a new generation of entrepreneurs, which is crucial for any economy to thrive sustainably. The creation of new entrepreneurs is aided by educational institutions, which work to boost students' intention to become entrepreneurs. The adoption of various digital technologies in the educational process has been prompted by the entry of Generation Z into the education system. The term "digital transformation" in this article describes how cutting-edge digital technologies, like online platforms, artificial intelligence, and interactive learning tools, are incorporated into entrepreneurial education, radically altering the way that knowledge is disseminated, accessible, and used. Technology adoption is only one aspect of it; it is part of a larger pedagogical movement that improves student engagement, fosters the development of digital skills, and builds learning settings that reflect the difficulties faced by entrepreneurs in the real world.

The deliberate willingness and planned commitment of an individual to engage in entrepreneurial activities, such as launching a business or pursuing self-employment, influenced by their own motivation, abilities, and surroundings, is referred to as entrepreneurial intention.

Although previous study has examined at the relationship between EE and EI, the results still remain mixed; some conclude that training has a favorable impact on intention, while others discover that the effects are weak or context-dependent [1].

The results of studies by Asimakopoulos et al., Cera et al., Iwu et al., and Wang et al. show a positive correlation between EE and the intention to pursue entrepreneurship [2–5]. Conversely, research by Reissova et al., Draksler and Sirec, and Martínez-Gregorio et al. casts doubt on or restricts the beneficial impact of EE on the intention to start a business [6–8]. According to Hattab, EE can foster EI by changing people's attitudes and thinking [9]. Duffy et al. identified macro factors such as the state of the economy and public policy, along with micro factors like personal resources, which are thought to help individuals maintain their ability to engage with their environment, manage situations, and significantly influence their career development [10]. Sweida & Sherman and Vamvaka et al. indicated that EI varies across genders [11,12]; Bergmann et al. emphasize the importance of spreading EE across non-business disciplines like IT and life sciences [13]. Furthermore, despite the integration of digital technologies into EE, little is known about how they specifically affect entrepreneurial intent, especially in non-Western European regions like Central and Eastern Europe (CEE) and Southern Europe (SE). Our knowledge of how digital transformation might successfully improve entrepreneurship outcomes in various educational environments is constrained by this gap.

                                                

This research contributes to theoretical knowledge and enhances our comprehension of the role of EE and digital solutions in fostering EI. To date, the application of digital technology in EE has not been extensively explored. In particular, the authors examine how 819 students in SE and CEE countries are affected by an AI-embedded digital tool compared to traditional workshop teaching. The study offers a multifaceted perspective on the elements influencing students' entrepreneurial intentions by incorporating both environmental and personal characteristics, including gender, prior business experience, education, and motivation, into the analysis.

The study provides empirical support for the usefulness of AI-enabled digital tools in entrepreneurship education, a field that has yet to be thoroughly studied, by concentrating on SE and CEE countries, it broadens the geographic reach of EE studies and emphasizes how contextual and cultural elements influence entrepreneurial intentions, moreover the study offers a thorough framework that enhances theory and practice by integrating digital change with environmental and human factors of emotional intelligence. The findings add to the continuing discussions about how higher education might encourage entrepreneurship in the digital age by showing that digital transformation at universities is a pedagogical driver of entrepreneurial capacity rather than just a technological innovation.

The study aims to investigate the impact of using a digital tool in EE on students' EI and the role of other environmental and personal factors in promoting EI based on a quasi-experiment conducted in Southern (SE) and Central and Eastern European (CEE) countries.

The limitations of the study are related to the sample of specific SE and CEE countries, which may limit generalizability to other regions, and the results are mainly based on self-reported data and one selected digital tool, as well as the quasi-experiment used in the study does not provide long-term insight into the development of students' entrepreneurial intention.

## Literature review

### Digital transformation and entrepreneurship education

The digital transformation has permeated various industries, including the education sector [14–16], and has significantly impacted how EE is delivered and perceived [17–19]. Integrating digital technologies, such as online learning platforms, virtual simulations, and collaborative tools, has revolutionized the learning experience for aspiring entrepreneurs [20,21].

EE has undergone a profound transformation in response to the digitalization of higher education. Integrating digital technologies, such as online learning platforms, virtual simulations, and collaborative tools, has not only reshaped pedagogical delivery but also enhanced the accessibility and effectiveness of EE [22]. These technologies foster immersive learning environments that simulate real-world entrepreneurial challenges, equipping students with practical competencies and digital fluency essential for modern entrepreneurial practice [23,24].

Moreover, digital platforms facilitate global collaboration, enabling students to engage with mentors and peers across borders, cultivating a broader entrepreneurial mindset [25]. As Angelova et al. emphasize, academic entrepreneurship increasingly relies on digital infrastructures to support innovation and knowledge transfer [26].

Recent studies also underscore the role of AI, simulation, and gamification in enhancing entrepreneurial competencies [27]. These tools improve engagement and support personalized learning pathways, critical for developing EI. Lambarri Villa et al. found that final-year undergraduate students attribute high importance to entrepreneurial competence, mainly when supported by digital learning environments [28]. Thus, digital transformation is not merely a technological shift but a pedagogical evolution directly influencing students' EI.

Digital technologies have a significant impact on EE in several ways:

- Application of new teaching techniques like simulations, virtual reality, and online platforms. These tools create immersive and engaging learning experiences for students [25];

- Enhanced accessibility: Online courses and resources make entrepreneurial education (EE) more available to a broader audience, regardless of their location or time limitations [29];

- Focus on digital skills – digital technologies necessitate a greater focus on digital skills in EE. Students need to learn how to leverage tools for marketing, e-commerce, data analysis, and more [30];

- Collaboration and networking: Online platforms and communication tools facilitate collaboration among students, mentors, and entrepreneurs worldwide, fostering a global entrepreneurial mindset [31].

Using digital tools, educational institutions can craft immersive and interactive learning environments that mimic real-world entrepreneurial scenarios. This method enables students to gain practical skills and firsthand experience in tackling entrepreneurial challenges. Arranz et al. examined the effects of digital transformation on entrepreneurial education (EE), emphasizing the crucial role of digital tools and technologies in shaping students' entrepreneurial intentions (EI). They discovered that incorporating digital technologies, including online platforms, data analytics, and Artificial Intelligence (AI), can enhance EE's effectiveness and increase students' EI [23].

Several groups of technologies, including simulation and gaming, AI and machine learning, and virtual worlds, positively impacted students' entrepreneurial competencies [24]. The study highlights the need for more empirical studies to examine the effects of educational technologies on entrepreneurial competencies, especially for newer technologies [32]. The authors suggest that future research should focus on validating the competency-based approach in EE outcomes assessment and gathering the views of entrepreneurship educators, managers, and educational technology experts. Integrating digital technologies into EE, focusing on the development of entrepreneurial skills, and fostering a supportive environment can greatly enhance students' EI.

## Entrepreneurship education and entrepreneurial intention

EI refers to the individual's readiness and willingness to engage in entrepreneurial activities, such as starting a new business or pursuing self-employment [33].

In the context of entrepreneurship education, digital transformation and entrepreneurial intention are intimately related. Although students' readiness to participate in entrepreneurial activities is reflected in their entrepreneurial intention, digital transformation offers the resources and settings that can mold this readiness. Universities can design learning experiences that boost self-esteem, motivation, and practical skills by combining artificial intelligence, internet platforms, and interactive simulations. In this way, digital transformation serves as a pedagogical catalyst that enhances students' entrepreneurial intention and equips them for success in digital economies.

As one of the most popular models for elucidating entrepreneurial intention (EI), Ajzen's Theory of Planned Behavior (TPB) [6,11,64] holds that entrepreneurial self-efficacy is closely related to intention, which is derived from attitudes toward entrepreneurship, perceived social norms, and perceived behavioral control [12,59]. TPB has been demonstrated to explain how training can impact EI in entrepreneurship education by influencing attitudes and boosting students' self-confidence in their entrepreneurial skills [6,51]. TPB is supplemented by Social Cognitive Theory (SCT) of Bandura, which highlights the importance of self-efficacy, vicarious learning, and reciprocal interaction between environmental and individual factors. These concepts are especially pertinent in educational settings where opportunities to strengthen entrepreneurial skills and motivation are provided by digital tools, peer collaboration, and experiential learning [46,53]. When combined, TPB and SCT provide a thorough framework for examining how students' entrepreneurial intention is influenced by their personal traits, educational interventions, and digital transformation.

A growing body of research has examined the factors that influence entrepreneurial intentions among students, with a particular focus on the role of EE [8,34–36]. Studies has shown that EE can positively influence students' EI [37]. By exposing students to practical aspects of entrepreneurship, such as business planning, opportunity recognition, and

risk management, EE can cultivate an entrepreneurial mindset and boost their confidence in pursuing entrepreneurial ventures.

EE is widely recognized as a catalyst for entrepreneurship [38].

The literature reveals numerous advantages of EE. The integration of practical skills, mentorship, and access to resources in EE programs has been identified as a key factor in shaping the EI of students. Research specifically indicates that EE has a greater impact on the intention to start a business compared to general business education [39]. This effect is due to the enhancement of students' perceived capabilities, knowledge, confidence, and access to resources and inspiration, which collectively foster a stronger inclination to establish a business. Studies show that participants in EE programs demonstrate increased abilities and aspirations to launch a business [2].

Cera et al. further explored the relationship between EE and EI, highlighting the importance of developing a comprehensive EE program that addresses cognitive and non-cognitive factors, such as creativity, risk-taking, and perseverance [3].

EE provides students with essential technical and managerial skills while fostering an entrepreneurial spirit, which can significantly drive EI [40]. By exposing students to real-world entrepreneurial challenges, case studies, and hands-on experiences, educational institutions can cultivate an entrepreneurial mindset and encourage students to view entrepreneurship as a viable career option.

Research has shown that EE programs that focus on developing practical skills, fostering a supportive environment, and providing access to entrepreneurial resources can significantly enhance students' EI [8].

Adeel et al. investigated the impact of EE on the EI of students. The study found that EE positively impacts students' EI by cultivating an entrepreneurial mindset, developing essential skills, and fostering a supportive environment for entrepreneurial activities [41].

In summary, the current literature strongly supports the notion that EE can significantly influence the EI of university students. The positive correlation between EE and EI has been noted across diverse cultural contexts, indicating that integrating EE into academic curricula can effectively encourage entrepreneurial activities and cultivate a culture of innovation among students. Additionally, personal characteristics and environmental factors, alongside EE and digital transformation, play a vital role in shaping students' EI.

## Personal and environmental factors influencing entrepreneurial intention

In addition to the role of EE, research has identified a range of personal and environmental factors that can influence an individual's EI. Personal traits, including self-efficacy, a propensity for risk-taking, and an entrepreneurial mindset, are key predictors of EI [42–44]. Individuals with strong self-belief in their abilities, a propensity for taking calculated risks, and an entrepreneurial mindset are more likely to pursue entrepreneurial ventures [45,46].

Environmental factors, such as family background, social networks, and access to entrepreneurial resources, can also shape an individual's EI [46,47]. Studies have shown that individuals from entrepreneurial families are more likely to develop an EI, as they may have been exposed to entrepreneurial role models and gained relevant knowledge and skills from an early age [3,48].

Liu et al. explored the effects of EE, prior entrepreneurial experience, and perceived entrepreneurial self-efficacy on EI among university students in the Netherlands. Their findings suggest that EE and prior entrepreneurial experience can enhance perceived entrepreneurial self-efficacy, positively influencing EI [49]. Studies by others, e.g., [50], have also shown similar results.

Martínez-Gregorio et al. investigated the relationship between EE and EI among university students in Spain. Their findings indicate that EE significantly influenced EI, and this relationship was mediated by entrepreneurial self-efficacy [8].

Mónico et al. examined the impact of EE, family business background, and personal traits on EI among university students in Portugal. Their study revealed that EE and family business background were positively associated with EI [51].

Studies have also found that the Big Five personality traits, gender differences, income levels, and age influence entrepreneurial intentions [52].

Mukhtar et al. explored the influence of EE, entrepreneurial self-efficacy, and entrepreneurial motivation on EI among university students in Indonesia. Their findings suggest that EE, entrepreneurial self-efficacy, and entrepreneurial motivation are significant predictors of EI [53].

Pinto Borges et al. studied the effect of EE on EI among university students in Portugal. Their results show that EE positively impacts EI, and this relationship is mediated by entrepreneurial self-efficacy [54].

## Research hypotheses

Entrepreneurial intention (EI) refers to an individual's conscious decision to pursue entrepreneurial activities, such as launching a business or engaging in self-employment [8]. EE is pivotal in shaping this intention by fostering an entrepreneurial mindset, enhancing self-efficacy, and providing exposure to real-world business scenarios [3,41]. Programs emphasizing practical skills, mentorship, and access to entrepreneurial resources have significantly boosted students' EI [39,54].

Based on the literature analysis, the hypothesis is stated as follows:

H1: There is a positive relationship between entrepreneurial training and EI.

In comparison to conventional teaching techniques, a number of studies highlight how using digital technologies into entrepreneurship education greatly increases students' entrepreneurial intention. AI-driven platforms, simulations, and interactive workshops are examples of digital technologies that offer immersive and customized learning experiences that enhance entrepreneurial motivation, self-efficacy, and opportunity recognition [24,27,37]. Additionally, studies have indicated that digital learning settings promote greater entrepreneurial intents by increasing engagement and more accurately simulating real-world entrepreneurial difficulties than traditional formats [22,28]. Based on these findings, it is logical to assume that training using the AI-based tool KABADA will result in a greater rise in entrepreneurial intention than that of conventional workshop techniques.

H2: The impact on EI after entrepreneurial training with the KABADA tool is more considerable than after the workshop.

Gender has been recognized as a factor in EI in several studies. Research shows that men and women may differ in their entrepreneurial motivations, risk tolerance, and responses to training interventions [55]. Atienza-Barba et al. found that ecological awareness impacts EI differently across genders, suggesting that gendered perceptions and values shape entrepreneurial pathways [56]. Studies have demonstrated that gender interacts with training formats, entrepreneurial passion, and cultural context to produce varied outcomes in EI [57,58]. Therefore, it is essential to consider gender as a demographic variable and a lens through which entrepreneurial training is experienced and internalized.

Previous studies support the following formulation of the hypothesis:

H3: The gender of entrepreneurial training participants impacts EI intensity.

Education level has long been considered a foundational factor influencing entrepreneurial intention (EI). Silesky-Gonzalez et al. found that while entrepreneurship education may not directly stimulate EI, it significantly enhances perceived behavioral control, one of the key antecedents in the Theory of Planned Behaviour [59]. Higher education levels may indirectly foster EI by improving students' confidence and perceived feasibility of entrepreneurial action. Le et al. further demonstrated that among master's students, education positively correlates with perceived desirability and feasibility, both of which mediate the relationship between education and EI [60]. These findings support the following hypothesis statement:

H4: The education levels of entrepreneurial training participants impact EI intensity.

Entrepreneurial knowledge, defined as the understanding of business creation processes, risk management, and opportunity recognition, is critical in shaping EI. Roxas et al. proposed a conceptual framework showing that knowledge gained from formal entrepreneurship courses positively affects EI by enhancing attitudes and social norms favourable to

entrepreneurship [61]. Pham et al. extended this by demonstrating that entrepreneurial knowledge, combined with technological innovativeness, strengthens students' motivation and perceived feasibility, boosting EI [62]. These studies validate the hypothesis formulation:

H5: There is a positive relationship between entrepreneurial training participants' knowledge of entrepreneurship self-assessment and EI.

Entrepreneurial experience, whether direct (e.g., starting a business) or vicarious (e.g., family background), has a profound impact on EI. Bozward and Rogers-Draycott found that both types of experience influence intention across short-, medium-, and long-term horizons, with family experience particularly linked to long-term aspirations. Giones et al. confirmed that prior entrepreneurial experience enhances EI through the mediating effects of personal attitude, social norms, and perceived behavioural control, as outlined in the TPB model [63]. These findings substantiate the hypothesis:

H6: There is a positive relationship between the experience in entrepreneurship of entrepreneurial training participants and EI.

While much of the literature focuses on cognitive and structural factors, motivational drivers are equally critical in shaping EI. Entrepreneurial motivation encompasses intrinsic passion, goal orientation, and the desire for autonomy, which are often cultivated through EE [55]. Dong and Bao highlight the role of affective events, such as exposure to entrepreneurial narratives, in triggering motivational shifts that lead to stronger EI [27]. Motivation acts as a catalyst that transforms knowledge and skills into actionable intent. Programs incorporating storytelling, experiential learning, and personalized goal setting are more likely to foster sustained entrepreneurial motivation [45,53].

Drawing upon insights from the reviewed literature, the following hypothesis is proposed:

H7: There is a positive relationship between the entrepreneurial training participant motivation and EI.

## Materials and methods

### Sample and analysis method

In the empirical part, the authors performed statistical data analysis based on the results of the quasi-experiment conducted in selected Southern (SE) and Central and Eastern European (CEE) countries.

The study focuses on sample groups from European regions with unique mentalities and recently transformed political and economic frameworks. The choice of location, especially Latvia, is due to the scarcity of research in these countries. The authors emphasize the need to understand the differences in entrepreneurship across various countries, as it is widely recognized as a catalyst for economic development. This is particularly important for countries shifting from planned to market economies.

A quasi-experimental method was used to investigate how the usage of a digital tool in entrepreneurship education affects Generation Z students' entrepreneurial intention. The type of educational workshop (AI-based digital tool KABADA vs. traditional workshop) was the independent variable in this study, and the students' intention to start their own business was the dependent variable. The KABADA tool was used to teach entrepreneurship to the experimental group (treatment group), whereas a traditional workshop without digital assistance was used to teach the identical material to the control group. Conditions were balanced because the two groups were similar in terms of geography, education, occupation, and other characteristics; the selection of the students can be considered random. Pre-test and post-test surveys, which had the same sets of questions before and after the workshops to gauge response changes, were used to gather data. This made it possible to directly compare the outcomes of traditional and digital education methods. A 7-point Likert scale was used to record responses, making it possible to evaluate changes in entrepreneurial intention. The design allowed for both exploratory insights into the direction and strength of the intervention's effects as well as confirmatory testing of hypotheses.

The research sample of 819 was collected from students who participated in the quasi-experiment, which was split into two groups: the experimental group (KABADA workshop) and the control group (conventional workshop). The age, gender, educational level, and previous entrepreneurial experience of the respondents were recorded both before and

after training. The surveys were conducted in accordance with the ethical principles described in the Ethics Statement of BA School of Business and Finance, and confirmations were obtained from the respondents that they did not object to the publication of the data.

The majority of participants were younger than 25. Students under the age of 22 made up 39.1% of the KABADA group prior to the intervention and 41.8% following it, compared to 52.0% and 50.6% in the typical workshop group. Students between the ages of 22 and 25 made up 26.5% and 32.2% of the traditional group, and 35.8% (before) and 32.9% (after) of the KABADA group. The KABADA group had 25.1% and 25.4% of students over 25, whereas the traditional group had 21.6% and 17.2% of students over 25.

Males made up 49.8% (before) and 52.1% (after) of the KABADA group and 48.0% and 48.3% of the conventional group, respectively, indicating a balanced gender distribution. In contrast to 52.0% and 51.7% in the traditional group, female students made up 50.2% (before) and 47.9% (after) of the KABADA group.

Students from bachelor's, master's, and doctorate programs were included in both groups based on their level of study. Bachelor's degree holders made up the largest percentage of participants, followed by master's degree holders and, finally, doctorate students.

Although a significant percentage of participants claimed some level of entrepreneurial activity or exposure, whether through self-employment, family businesses, or internships, the distribution of participants with regard to entrepreneurial experience showed that the majority had no prior entrepreneurial background. Both groups' self-reported entrepreneurship experience slightly increased following the interventions.

Ordinal logistic regression (OLR) analysis was chosen due to the purpose of research, and data specific – variable values (independent variables – gender, education level, experience, knowledge, interest, etc.; moderator variable- digital tool) were collected using Likert (1–7) scale, were categorical and ordered. This regression model has proven itself in many similar studies [6,64–66].

## Variables and regression model

The authors determined the following independent variables: gender (GEND), education (EDUC), experience in entrepreneurship (EXPE), knowledge of entrepreneurship self-assessment (KNSA), entrepreneurship could fulfil your life (ESFL), entrepreneurship interests me (ESIT), interest in becoming an entrepreneur (IINT), education tool (TOOL), and education training: pre-post (EDTR). The intention to become an entrepreneur (INTE) was selected as the dependent variable.

The model under investigation:

$$INTE = f(x_{TOOL}, x_{EDTR}, x_{GEND}, x_{EDUC}, x_{EXPE}, x_{KNSA}, x_{ESFL}, x_{ESIT}, x_{IINT}) \tag{1}$$

The literature reviewed provides a basis for the validity of the questionnaire. Additionally, the internal consistency of the questionnaire was confirmed using Cronbach's alpha, exceeding the values by 0.7, demonstrating an adequate level of reliability.

The metric for evaluating a construct's convergent validity is the average variance extracted (*AVE*) for all variables. The minimum acceptable *AVE* is 0.50 – an *AVE* indicates the variance of the indicators that make up the construct. The *AVE* values (min 0.526) are above the required minimum level of 0.50, thus showing an acceptable level of convergent validity.

The questionnaire for this study was designed based on Aizen's Theory of Planned Behaviour and related studies [6,11,64], etc., thus providing substantiation for content validity.

For construct validity evaluation, a Spearman correlation analysis was used.

As shown in Table 1, all independent variables' correlations are statistically significant at a confidence level > 95% with one exception – education level. Despite the lack of statistical significance, the independent variable EDUC was not excluded from further analysis in order to test hypothesis H4.

**Table 1. Spearman correlation analysis results.**

|  | Spearman's rho | *p*-value | Significance |
|---|---|---|---|
| EDTR | 0.0868 | 0.0130 | yes |
| TOOL | 0.1031 | 0.0031 | yes |
| GEND | 0.1506 | <0.0001 | yes |
| EDUC | −0.0077 | 0.8265 | no |
| KNSA | 0.4211 | <0.0001 | yes |
| EXPE | 0.3129 | <0.0001 | yes |
| ESFL | 0.6102 | <0.0001 | yes |
| ESIT | 0.7186 | <0.0001 | yes |
| IINT | 0.6498 | <0.0001 | yes |

The internal consistency of the questionnaire was validated by applying Cronbach's alpha. As the actual alpha value (0.81) exceeds the threshold of 0.7, it demonstrates an adequate level of reliability.

The metric for evaluating a construct's convergent validity is the average variance extracted (*AVE*) for all variables. The minimum acceptable *AVE* is 0.50 – an *AVE* indicates the indicators' variance that make up the construct. The *AVE* values (min 0.526) are above the required minimum level of 0.50 and thus show an acceptable level of convergent validity.

# Results

## OLR model and parameters

OLR model under calibration can be defined as follows:

$$\Pr\left\{INTE \leq c\right\} = \frac{e^t}{1 + e^t}$$

(2)

$$\text{were } t = \beta_0 + \beta_1 * x_{TOOL} + \beta_2 * x_{EDTR} + \beta_3 * x_{GEND} + \beta_4 * x_{EDUC} + \beta_5 * x_{EXPE}$$
$$+ \beta_6 * x_{KNSA} + \beta_7 * x_{ESFL} + \beta_8 * x_{ESIT} + \beta_9 * x_{IINT}$$

(3)

$\beta_0$ – intercept,

$\beta_1 \ldots \beta_9$ – regression coefficients,

$x_i$ – regression variables.

Calibrated OLR model statistics are Likelihood ratio statistic (791.2) and corresponding *p*-value (<2.2e-16), allowing the conclusion that the calibrated OLR model is statistically stable at a confidence level higher than 99% and can be used for interpretation. Pseudo *R* squared, according to Nagelkerke (Cragg and Uhler) (0.6367) and Cox and Snell (ML) (0.6194), allows the conclusion that with variables included in the OLR model, one can explain more than 61% of variations in the assessment of dependent variable "Intention to become an entrepreneur" [66].

The following Table 2 summarizes the statistics of calibrated logistic regression coefficients.

As one can see, all with one exception (EDUC) regression coefficients are statistically significant at a confidence level of 95%, thus supporting hypotheses H1 – H3 and H5 – H7.

Furthermore, regression coefficient values allow us to conclude that the most influential EI factors are ESIT (0.7570), IINT (0.4703), EDTR (0.2724), EXPE (0.3392), and ESFL (0.3234).

Table 3 summarizes the statistics of the calibrated logistic regression model intercepts. All statistics are statistically significant at a high confidence level of >99%.

**Table 2. The statistics of calibrated logistic regression coefficients and hypotheses test results.**

| Variable | $\widehat{\beta_i}$ | Std.Error | z-value | p-value | Signif. | 2.5% LCI | 97.5% UCI | Hypotheses |
|---|---|---|---|---|---|---|---|---|
| EDTR | 0.3596 | 0.1386 | 2.5938 | 0.0095 | ** | 0.0882 | 0.6318 | H1 sup |
| TOOL | 0.2724 | 0.1366 | 1.9942 | 0.0461 | * | 0.0049 | 0.5405 | H2 sup |
| GEND | 0.2748 | 0.1349 | 2.0369 | 0.0417 | * | 0.0106 | 0.5396 | H3 sup |
| EDUC | 0.0679 | 0.0827 | 0.8209 | 0.4117 | | −0.0940 | 0.2305 | H4 not sup |
| KNSA | 0.2449 | 0.0612 | 4.0011 | 0.0000 | *** | 0.1252 | 0.3653 | H5 sup |
| EXPE | 0.3392 | 0.0926 | 3.6656 | 0.0002 | *** | 0.1584 | 0.5213 | H6 sup |
| ESFL | 0.3234 | 0.0722 | 4.4798 | 0.0000 | *** | 0.1826 | 0.4657 | H7 sup |
| ESIT | 0.7570 | 0.0820 | 9.2328 | 0.0000 | *** | 0.5974 | 0.9190 | H7 sup |
| IINT | 0.4703 | 0.0719 | 6.5399 | 0.0000 | *** | 0.3301 | 0.6121 | H7 sup |

Signif. codes: 0 '***' 0.001 '**' 0.01 '*' 0.05 '.' 0.1 ' ' 1.

**Table 3. Calibrated logistic regression model intercepts statistics.**

| Intercept | $\widehat{\beta_0}$ | Std.Error | z-value | p-value | Signif. |
|---|---|---|---|---|---|
| INTE 1\|2 | 3.8145 | 0.5819 | 6.5546 | 0.0000 | *** |
| INTE 2\|3 | 5.5805 | 0.5710 | 9.7740 | 0.0000 | *** |
| INTE 3\|4 | 7.1250 | 0.5876 | 12.1258 | 0.0000 | *** |
| INTE 4\|5 | 8.8842 | 0.6189 | 14.3540 | 0.0000 | *** |
| INTE 5\|6 | 10.5831 | 0.6485 | 16.3194 | 0.0000 | *** |
| INTE 6\|7 | 12.7568 | 0.6859 | 18.5985 | 0.0000 | *** |

Signif. codes: 0 '***'.

The regression coefficients estimated in Table 3 are scaled in terms of logs and, therefore, are difficult to interpret. Table 4 summarizes the odds ratios of the corresponding statistics.

Odds ratios (see Table 4) allow us to provide the following interpretation: 1) entrepreneur training participants after KABADA have 1.31 times higher EI than after the workshop, given that the other variables in the model (2) are held constant; 2) male students have 1.32 time higher EI then females, given that the other variables in the model (2) are held constant; 3) for one grade increase in students self-assessment on question "Entrepreneurship interests me" the EI self-assessment increases 2.13 times, given that the other variables in the model (2) are held constant etc.

**Table 4. Odds ratios of corresponding statistics.**

| Variable | $\exp(\widehat{\beta_i})$ | 2.5% LCI | 97.5% UCI |
|---|---|---|---|
| EDTR | 1.4328 | 1.0922 | 1.8810 |
| TOOL | 1.3131 | 1.0049 | 1.7169 |
| GEND | 1.3163 | 1.0107 | 1.7154 |
| EDUC | 1.0703 | 0.9102 | 1.2592 |
| KNSA | 1.2775 | 1.1334 | 1.4409 |
| EXPE | 1.4039 | 1.1716 | 1.6843 |
| ESFL | 1.3817 | 1.2003 | 1.5932 |
| ESIT | 2.1318 | 1.8175 | 2.5067 |
| IINT | 1.6004 | 1.3911 | 1.8442 |

## OLR model validation

The following OLR assumptions were made:

1. The dependent variable values are ordered;

2. Independent variables are either continuous, categorical, or ordinal;

3. Multi-collinearity is not present;

4. Proportional odds hold.

The dataset used to calibrate the OLR model satisfied assumptions 1 and 2. To check the multi-collinearity assumption, the authors use the variance inflation (*if*) test (see following Table 5).

As *vif* statistics for all OLR model independent variables are below 3, the authors can conclude that there is no evidence of multi-collinearity in the dataset used for calibration of model 2, and assumption tree is met.

The Brant test was employed to check the assumption on proportional odds, which means that the relationship between each pair of outcome groups must be the same.

As Brant test *p*-values for all variables are > 0.05 (see Table 6), the authors conclude that the proportion odds assumption holds.

The study's results revealed that the digital tool KABADA significantly impacts EI in the learning process, thus confirming the findings that digitization-based EE is effective in promoting EI.

The empirical section's sampling and analytical methods ensure the study's external validity and generalizability, suggesting that digital tools can effectively engage diverse student demographics. Interest in entrepreneurship, desire to be an entrepreneur, educational training before starting, experience in entrepreneurship, and entrepreneurship are considered lifestyle influences of EI. Therefore, personal characteristics are essential.

## Discussion

This study provides compelling evidence that both personal factors and education, particularly when enhanced by digital tools, significantly influence students' EI. The findings reinforce the growing consensus in EE literature that motivation, self-assessment, and experiential learning are central to fostering EI among university students [3,41].

The strongest predictors of EI, entrepreneurship interests me (ESIT), interest in becoming an entrepreneur (IINT), and the belief that entrepreneurship could fulfil one's life (ESFL), highlight the motivational and affective dimensions of entrepreneurial intention. These results align with Dong and Bao, who emphasize the role of affective events and personal narratives in shaping entrepreneurial motivation [27]. Similarly, Mukhtar et al. and Maheshwari et al. argue that intrinsic motivation acts as a catalyst that transforms knowledge into entrepreneurial action [45,53].

The significant impact of prior entrepreneurial experience (EXPE) and self-assessed entrepreneurial knowledge (KNSA) supports earlier findings by Bozward and Rogers-Draycott and Roxas et al., who noted that experiential learning

**Table 5. The results of the variance inflation (*vif*) test.**

| Variable | TOOL | EDTR | GEND | EDUC | KNSA | EXPE | ESFL | ESIT | IINT |
|---|---|---|---|---|---|---|---|---|---|
| *vif* | 1.0910 | 1.1055 | 1.0848 | 1.6448 | 1.6936 | 1.3988 | 2.0236 | 2.8980 | 2.8957 |

**Table 6. The results of the Brant test.**

| Variable | TOOL | EDTR | GEND | EDUC | KNSA | EXPE | ESFL | ESIT | IINT |
|---|---|---|---|---|---|---|---|---|---|
| *p*-value | 0.09 | 0.08 | 0.50 | 0.24 | 0.23 | 0.13 | 0.07 | 0.33 | 0.07 |

and self-efficacy are critical in shaping long-term entrepreneurial aspirations [61]. These results suggest that EE programs should incorporate experiential components and self-assessment tools to enhance students' confidence and perceived feasibility of entrepreneurial endeavours.

Gender also emerged as a significant factor, with male students showing higher EI than female counterparts. This finding is consistent with studies by Sweida & Sherman, Vamvaka et al., and Gomes et al., which highlight gender-based differences in entrepreneurial motivation and risk tolerance [11,12,57,67,68]. The implication is that EE should adopt gender-sensitive pedagogical strategies to ensure inclusivity and effectiveness across diverse student populations.

Interestingly, education level did not significantly predict EI, echoing the mixed results found in prior research [6,59]. While higher education may enhance perceived behavioural control, it does not necessarily translate into stronger entrepreneurial intention [60]. This suggests that the content and delivery of EE, rather than the level of education, are more critical in shaping entrepreneurial outcomes.

The use of the AI-based digital tool KABADA and the pre-post training format significantly enhanced EI, validating the hypothesis that digital transformation in EE can serve as a pedagogical driver rather than merely a technological upgrade. These findings are in line with Arranz et al., Hammoda, and Sebastián-Rivera et al., who argue that immersive digital environments foster practical competencies and entrepreneurial fluency [22,23,28]. Moreover, the integration of AI and interactive platforms supports personalized learning pathways, which are essential for cultivating entrepreneurial skills [27,28].

Overall, the study underscores the importance of combining digital tools with motivational and experiential learning strategies to enhance EI. It also highlights the need for EE programs to be context-sensitive, especially in regions like Southern and Central-Eastern Europe, where cultural and economic factors may shape entrepreneurial aspirations differently [25,37].

## Conclusion

According to this study, university students' entrepreneurial intention is strengthened by digital transformation. The research investigated how interactive and technology-driven learning settings significantly improve students' motivation, self-assessment, and preparedness to launch a business by contrasting traditional workshops with an AI-based digital tool. While education level by itself did not predict intention, gender and previous entrepreneurial experience did have an impact.

This suggests that personal traits and practical participation are more important than formal educational background. These findings lend credence to the idea that digital transformation in higher education is a pedagogical factor that fosters entrepreneurial capacity in addition to being a technology change. In order to educate students for success in digital economies, the findings highlight the necessity for educators and policymakers to incorporate digital tools and skill-building into entrepreneurship education.

The findings demonstrate that EE workshops utilizing the digital tool KABADA positively impact students' EI across various educational levels and fields, both for business and non-business students.

With AI increasingly integrating into the educational sector, including EE, the authors advocate for leveraging AI's potential in higher education institutions. The study's outcomes support the optimistic view of incorporating AI in digital educational tools for EE. Nevertheless, given the nascent stage of AI application in EE, the authors urge researchers to devote more attention to investigating AI's potential to enhance EE quality and the risks associated with adopting unverified AI solutions in educational settings.

Compared to traditional workshops, AI-based and interactive digital technologies greatly increase students' entrepreneurial intention, so universities should incorporate them into entrepreneurship education.Since these have a significant impact on entrepreneurial outcomes, educators should concentrate on individual variables like motivation, past business experience, and gender-sensitive approaches. Higher education's digital transformation should be viewed as both a pedagogical approach to preparing students for entrepreneurship in a digital economy and a technological advancement.

By encouraging the creation and uptake of digital platforms that boost students' self-esteem, motivation, and confidence, policymakers can fortify entrepreneurial ecosystems.

Recommendations for further research are based on the limitations of the study and involve conducting longitudinal studies to investigate whether educational training with digital tools influences the effectiveness of students' entrepreneurial decision-making processes in real-world business, investigating gender differences in the implementation of these processes, conducting cross-cultural comparisons outside of selected regions, and finally investigating how AI influences the promotion of entrepreneurial intentions by comparing several AI-based digital tools.

## Author contributions

**Conceptualization:** Aivars Spilbergs, Inese Mavlutova, Kristaps Lesinskis.

**Data curation:** Aivars Spilbergs, Kristaps Lesinskis.

**Formal analysis:** Aivars Spilbergs, Inese Mavlutova.

**Funding acquisition:** Kristaps Lesinskis.

**Investigation:** Aivars Spilbergs, Kristaps Lesinskis.

**Methodology:** Aivars Spilbergs, Inese Mavlutova.

**Project administration:** Inese Mavlutova, Kristaps Lesinskis.

**Software:** Aivars Spilbergs.

**Supervision:** Inese Mavlutova.

**Validation:** Aivars Spilbergs.

**Writing – original draft:** Aivars Spilbergs.

**Writing – review & editing:** Aivars Spilbergs, Inese Mavlutova.

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
