## [Decision Letter · Decision Letter 0]

30 Jul 2025

PONE-D-25-05271Entrepreneurial Intention of Students: The Role of Digital Transformation and Use of Digital Tools in Education and Other Environmental and Personal FactorsPLOS ONE

Dear Dr. Lešinskis,

Thank you for submitting your manuscript to PLOS ONE. After careful consideration, we feel that it has merit but does not fully meet PLOS ONE’s publication criteria as it currently stands. Therefore, we invite you to submit a revised version of the manuscript that addresses the points raised during the review process.

 Please submit your revised manuscript by **Sep 13 2025 11:59PM** . If you will need more time than this to complete your revisions, please reply to this message or contact the journal office at plosone@plos.org . Please include the following items when submitting your revised manuscript:

We look forward to receiving your revised manuscript.

Kind regards,

Ali Junaid Khan, PhD

Academic Editor

PLOS ONE

Journal Requirements:

“The research is financed by the Recovery and Resilience Facility project "Internal and External Consolidation of the University of Latvia" (No.5.2.1.1.i.0/2/24/I/CFLA/007)”

4. Please note that funding information should not appear in the Acknowledgments section or other areas of your manuscript. We will only publish funding information present in the Funding Statement section of the online submission form. Please remove any funding-related text from the manuscript. 

5. In the online submission form, you indicated that: “Available in Argos Openaire platform.”

**Additional Editor Comments:**

Be sure to:

Rewrite your abstract and follow the IMRAD approach (Introduction, Method, Results, and Discussion)Improve introduction and clearly explain the gap and problem of study.

Reviewers' comments:

Reviewer's Responses to Questions

**Comments to the Author**

1. Is the manuscript technically sound, and do the data support the conclusions?

Reviewer #1: Partly

Reviewer #2: Partly

2. Has the statistical analysis been performed appropriately and rigorously? 

Reviewer #1: No

Reviewer #2: Yes

3. Have the authors made all data underlying the findings in their manuscript fully available?

Reviewer #1: No

Reviewer #2: Yes

4. Is the manuscript presented in an intelligible fashion and written in standard English?

Reviewer #1: Yes

Reviewer #2: Yes

5. Review Comments to the Author

Reviewer #1: This paper addresses an up-to-date and interesting topic. This paper aims to investigate the impact of the digital transformation and digital tools in entrepreneurship education and their role in entrepreneurial intentions. The manuscript has a great potential, but reasoning does not flow naturally.

Although the idea underlying the work is interesting, I consider that the article must be improved in line with the comments below:

1. Please make your abstract attractive to readers (simple sentences without any repetition) and include 2-3 sentences ready to be cited exactly as they are. In one paragraph, your abstract should tell the readers why the study is important (maximum 25% of the text), what you did, i.e. your methodology (maximum 25% of the text), and what you found, i.e. main research results and their major implications (50% of the text). This is very important to promote your work because of the growing trend that authors use Google search to find and cite papers based on the abstract (instead of reading the full paper).

2. The introduction does not clearly argue with the interest of the research purpose. The first two paragraphs need to be enhanced with more arguments exemplifying the importance of entrepreneurship and this research. There are a lot of published work in 2025, such as Sebastian-Rivera et al. (2025, DOI: 10.1080/10447318.2025.2474489) or Atienza-Barba et al. (2025, DOI: 10.1007/s11365-025-01084-7), to strengthen the narrative on this. I miss a paragraph where you make it clear what are the contributions of your work to the literature.

3. The study aims to contribute to a large body of existing literature on digital transformation topics in the context of education. A new explanation is interesting only if it tells us things that other explanations cannot. What is the theoretical approach of this paper? Please see Barba-Sánchez et al. (2022, DOI: 10.3145/epi.2022.nov.17).

4. I require that you add text for each one of your hypotheses and justify each one of them separately and in a better manner you do now. Please try to separate the particular text that corresponds with one or another hypothesis.

5. An article must be self-contained, i.e. it must contain all the information necessary for its comprehension. How the sample has been chosen and to demonstrate that it is representative of the society being studied. I would also recommend adding a table with accurate information about the technical specifications of the study and another table with the comparative demographics between the population and sample.

6. In the methodology, the authors did not properly present the process of using "quasi-experiment" as a tool for gathering information.

7. It would be interesting to explain how the proposed model variables are measured. Add an annexed with the items that measure the variables.

8. Please add the validity and reliability of the measurement instruments. It is not enough to report the Cronbach's alpha coefficient and the average variance extracted (AVE).

9. The paper would be more interesting in more sophisticated analysis. If the questionnaire permits, which seems possible although do not have a copy of it in research, other technique would be used to test the hypotheses, such as Structural Equations Models (SEM) using the technique of Partial Least Square (PLS). Why did you not use SEM to test the hypotheses?

10. The Conclusion section does not offer value to the reader. The logic behind building research relevance should be sharpened. The authors should develop more the theoretical and practical implications of the study for the universities, teachers, and students/society.

11. Where are the limitations of this work? Please add a heading specifically for it.

I hope you find the above comments useful and I wish you the best of luck with developing the paper further. Finally, I'd like to compliment the authors for having made serious and interesting efforts in developing new approaches in corporate entrepreneurship.

Reviewer #2: Abstract

Basically, the abstract does not convey the appropriate and relevant information for its readers. The key findings and methodology are obviously missing in the abstract and this does not allow the reader to appreciate the work done. The abstract has no recommendations as well as practical implications. These two acronyms “CEE and SE” are used for the first time without their full meanings. The use of other environmental and personal factors in the topic is confusing. Which factors are classified as other factors?

Introduction

The introduction somehow attempts to provide relevant and appropriate information on the subject area. However, the researchers need to focus on the following: In the first paragraph under line 6, the authors tried to define digital transformation as, "digital transformation" refers to the difficulty of adjusting to digital technology and has been chiefly examined as a process that occurs in several industries, such as mechanical engineering and finance. This definition is misleading. Also, the way and manner the authors cite scholars in the text is not appropriate and consistent. For eg, under line three of the second paragraph, “The results of studies by Asimakopoulos et al., Cera et al., Iwu et al., and Wang et al. show a positive correlation between EE and the intention to pursue entrepreneurship [3, 4, 5, 6]”. The numbers 3, 4,5 and 6 should not be at the end of the sentence. Again, the citations in this sentence “Conversely, research by Reissová et al., Draksler and Sirec, and Martínez-Gregorio et al. casts doubt on or restricts the beneficial impact of EE on the intention to start a business” are not complete.

The introduction does not provide information on how the use of digital tools in education and digital transformation enhance entrepreneurial intentions but rather focus on how education enhances entrepreneurial intention. It is not clear from the introduction how digital transformation and the use of digital tools in education drives intentions.

Literature Review

Essentially, the authors have demonstrated some level of understanding by reviewing relevant and appropriate literature in the areas of entrepreneurial intention, digital transformation, entrepreneurial education, personal and environmental factors. However, the authors need to focus on the following:

1. The authors should rather use transitional words in a paragraph to review the impact of digital technologies on EE instead of bulletins.

2. The authors need to include the theory underpinning the study.

3. The authors need to talk about the workshop that they are referring to in the second hypothesis. The readers do not know anything about the workshop and can not relate to what you talking about

4. The hypothesis 3(H3) talks about the gender of entrepreneurial training participants yet you did not discuss about that training in anywhere. Also, there is no review of literature on gender and entrepreneurial intentions enhance what forms the basis of this hypothesis?

5. What exactly do the researchers mean in the H4 by saying the education of entrepreneurial training participants? Are the looking at the educational level or what exactly?

6. Again, there is no literature review to support H7 so it becomes difficult to understand what necessitated the formulation of the hypothesis.

Materials and methods

How the researchers determined and selected the sample size of 819 students is not clear. How many students participated in the workshop and how many were selected and on what basis? How many were from the Southern European countries and how many were from the Central and Eastern European countries? So, what was the criteria for inclusion and exclusion? How many were selected from the control group and how many from the experimental group? What were the justifications for the numbers chosen?

Moreover, how did the authors collect the data from the respondents?

How the researchers measure both the dependent and independent variables are not known. Also, how they measure the control variables has not been stated.

Furthermore, the authors need to include a table of the validity and reliability measures.

Results

The results are appropriate and relevant

Discussions

The discussions have nothing to do with entrepreneurial intentions of students but rather focus on how the use of digital technologies enhanced academic efficiency. It is therefore, difficulty to see how the findings are related to the researchers’ topic (entrepreneurial intention of students). The discussion is limited in context and does not address the findings of the study. The authors need to discuss their findings and relate that to previous studies as well as discuss the practical and theoretical implications of their findings

Conclusion

The conclusions are not in line with the findings. This is because the authors did not discuss their findings.

References

The authors have cited relevant authors. However, they to need to cite more current research.

6. PLOS authors have the option to publish the peer review history of their article (what does this mean? ). If published, this will include your full peer review and any attached files.

**Do you want your identity to be public for this peer review?** For information about this choice, including consent withdrawal, please see our Privacy Policy .

Reviewer #1: No

Reviewer #2: No

---

## [Author Response · Author response to Decision Letter 1]

30 Sep 2025

Dear reviewers,

Herewith we provide our answers regarding your comments for the paper “Entrepreneurial Intention of Students: The Role of Digital Transformation and Use of Digital Tools in Education and Other Environmental and Personal Factors” to the journal PLOS ONE.

All journal requirements that apply to us have been taken into account and met.

Our comments regarding reviewer comments:

First of all, thank you for your comments, we tried to take them into account as much as possible and make appropriate improvements.

1. Please make your abstract attractive to readers (simple sentences without any repetition) and include 2-3 sentences ready to be cited exactly as they are. In one paragraph, your abstract should tell the readers why the study is important (maximum 25% of the text), what you did, i.e. your methodology (maximum 25% of the text), and what you found, i.e. main research results and their major implications (50% of the text). This is very important to promote your work because of the growing trend that authors use Google search to find and cite papers based on the abstract (instead of reading the full paper).

Abstract has been revised according to your suggestions.

2. The introduction does not clearly argue with the interest of the research purpose. The first two paragraphs need to be enhanced with more arguments exemplifying the importance of entrepreneurship and this research. There are a lot of published work in 2025, such as Sebastian-Rivera et al. (2025, DOI: 10.1080/10447318.2025.2474489) or Atienza-Barba et al. (2025, DOI: 10.1007/s11365-025-01084-7), to strengthen the narrative on this. I miss a paragraph where you make it clear what are the contributions of your work to the literature.

Requested improvements made to the introduction, paragraph where authors make it clear what are the contributions of our work to the literature added.

3. The study aims to contribute to a large body of existing literature on digital transformation topics in the context of education. A new explanation is interesting only if it tells us things that other explanations cannot. What is the theoretical approach of this paper? Please see Barba-Sánchez et al. (2022, DOI: 10.3145/epi.2022.nov.17).

The revised article includes references to relevant theories, such as Ajzen's Theory of Planned Behavior and others.

4. I require that you add text for each one of your hypotheses and justify each one of them separately and in a better manner you do now. Please try to separate the particular text that corresponds with one or another hypothesis.

We have taken note and it has been changed.

5. An article must be self-contained, i.e. it must contain all the information necessary for its comprehension. How the sample has been chosen and to demonstrate that it is representative of the society being studied. I would also recommend adding a table with accurate information about the technical specifications of the study and another table with the comparative demographics between the population and sample.

All the information requested in this comment has been included and described in the new version.

6. In the methodology, the authors did not properly present the process of using "quasi-experiment" as a tool for gathering information.

Now it is presented.

7. It would be interesting to explain how the proposed model variables are measured. Add an annexed with the items that measure the variables.

In the revised version we explain how the proposed model variables are measured.

8. Please add the validity and reliability of the measurement instruments. It is not enough to report the Cronbach's alpha coefficient and the average variance extracted (AVE).

Validity and reliability of the measurement instruments added in the revised version.

9. The paper would be more interesting in more sophisticated analysis. If the questionnaire permits, which seems possible although do not have a copy of it in research, other technique would be used to test the hypotheses, such as Structural Equations Models (SEM) using the technique of Partial Least Square (PLS). Why did you not use SEM to test the hypotheses?

We agree that the use of SEM and PLS would be useful, however, in this study we used regression analysis as a rather appropriate method. However, we plan to use SEM and PLS in future studies.

10. The Conclusion section does not offer value to the reader. The logic behind building research relevance should be sharpened. The authors should develop more the theoretical and practical implications of the study for the universities, teachers, and students/society.

Significant changes have been made to the conclusions section, including taking into account the above recommendations.

11. Where are the limitations of this work? Please add a heading specifically for it.

I hope you find the above comments useful and I wish you the best of luck with developing the paper further. Finally, I'd like to compliment the authors for having made serious and interesting efforts in developing new approaches in corporate entrepreneurship.

Limitations described in the revised version.

Reviewer #2:

Abstract

Basically, the abstract does not convey the appropriate and relevant information for its readers. The key findings and methodology are obviously missing in the abstract and this does not allow the reader to appreciate the work done. The abstract has no recommendations as well as practical implications. These two acronyms “CEE and SE” are used for the first time without their full meanings. The use of other environmental and personal factors in the topic is confusing. Which factors are classified as other factors?

Abstract has been revised according to your suggestions. Regarding the comment “The use of other environmental and personal factors in the topic is confusing. Which factors are classified as other factors?” we have changed the title of the paper and respectively approach that was misleading in the previous version.

Introduction

The introduction somehow attempts to provide relevant and appropriate information on the subject area. However, the researchers need to focus on the following: In the first paragraph under line 6, the authors tried to define digital transformation as, "digital transformation" refers to the difficulty of adjusting to digital technology and has been chiefly examined as a process that occurs in several industries, such as mechanical engineering and finance. This definition is misleading. Also, the way and manner the authors cite scholars in the text is not appropriate and consistent. For eg, under line three of the second paragraph, “The results of studies by Asimakopoulos et al., Cera et al., Iwu et al., and Wang et al. show a positive correlation between EE and the intention to pursue entrepreneurship [3, 4, 5, 6]”. The numbers 3, 4,5 and 6 should not be at the end of the sentence. Again, the citations in this sentence “Conversely, research by Reissová et al., Draksler and Sirec, and Martínez-Gregorio et al. casts doubt on or restricts the beneficial impact of EE on the intention to start a business” are not complete.

The introduction does not provide information on how the use of digital tools in education and digital transformation enhance entrepreneurial intentions but rather focus on how education enhances entrepreneurial intention. It is not clear from the introduction how digital transformation and the use of digital tools in education drives intentions.

Suggestions have been taken into account, and the appropriate explanations have also been included.

Literature Review

Essentially, the authors have demonstrated some level of understanding by reviewing relevant and appropriate literature in the areas of entrepreneurial intention, digital transformation, entrepreneurial education, personal and environmental factors. However, the authors need to focus on the following:

1. The authors should rather use transitional words in a paragraph to review the impact of digital technologies on EE instead of bulletins.

We have taken this recommendation into account and made the appropriate changes.

2. The authors need to include the theory underpinning the study.

The revised article includes references to relevant theories, such as Ajzen's Theory of Planned Behavior and others.

3. The authors need to talk about the workshop that they are referring to in the second hypothesis. The readers do not know anything about the workshop and can not relate to what you talking about.

We have added explanations about the workshops and their process in the new version.

4. The hypothesis 3(H3) talks about the gender of entrepreneurial training participants yet you did not discuss about that training in anywhere. Also, there is no review of literature on gender and entrepreneurial intentions enhance what forms the basis of this hypothesis?

The recommendations have been taken into account, and corrections have been made to the work to eliminate these shortcomings.

5. What exactly do the researchers mean in the H4 by saying the education of entrepreneurial training participants? Are the looking at the educational level or what exactly?

We have provided an explanation in the new version of the article.

6. Again, there is no literature review to support H7 so it becomes difficult to understand what necessitated the formulation of the hypothesis.

In the new version of the article, this problem has been resolved and explanations have been provided.

Materials and methods

How the researchers determined and selected the sample size of 819 students is not clear. How many students participated in the workshop and how many were selected and on what basis? How many were from the Southern European countries and how many were from the Central and Eastern European countries? So, what was the criteria for inclusion and exclusion? How many were selected from the control group and how many from the experimental group? What were the justifications for the numbers chosen?

Moreover, how did the authors collect the data from the respondents?

How the researchers measure both the dependent and independent variables are not known. Also, how they measure the control variables has not been stated.

Furthermore, the authors need to include a table of the validity and reliability measures.

All of the above comments have been taken into account, and explanations have been provided in the revised version of the article.

Results

The results are appropriate and relevant.

Discussions

The discussions have nothing to do with entrepreneurial intentions of students but rather focus on how the use of digital technologies enhanced academic efficiency. It is therefore, difficulty to see how the findings are related to the researchers’ topic (entrepreneurial intention of students). The discussion is limited in context and does not address the findings of the study. The authors need to discuss their findings and relate that to previous studies as well as discuss the practical and theoretical implications of their findings

We have significantly improved the discussion section based on your feedback.

Conclusion

The conclusions are not in line with the findings. This is because the authors did not discuss their findings.

Significant changes have been made to the conclusions section, including taking into account the above recommendations.

References

The authors have cited relevant authors. However, they to need to cite more current research.

Compared to the previous version, we have used a number of newer sources in the article and added them to the bibliography.

On behalft of our team of researchers,

Kristaps Lešinskis

---

## [Decision Letter · Decision Letter 1]

13 Nov 2025

Entrepreneurial Intention of Students: The Role of Digital Tools and Personal Factors in Entrepreneurship Education

PONE-D-25-05271R1

Dear Dr. Lešinskis

We’re pleased to inform you that your manuscript has been judged scientifically suitable for publication and will be formally accepted for publication once it meets all outstanding technical requirements.

Kind regards,

Ali Junaid Khan, PhD

Academic Editor

PLOS ONE

Additional Editor Comments (optional):

Reviewers' comments:

Reviewer's Responses to Questions

**Comments to the Author**

1. If the authors have adequately addressed your comments raised in a previous round of review and you feel that this manuscript is now acceptable for publication, you may indicate that here to bypass the “Comments to the Author” section, enter your conflict of interest statement in the “Confidential to Editor” section, and submit your "Accept" recommendation.

Reviewer #1: All comments have been addressed

2. Is the manuscript technically sound, and do the data support the conclusions?

Reviewer #1: Yes

3. Has the statistical analysis been performed appropriately and rigorously? 

Reviewer #1: Yes

4. Have the authors made all data underlying the findings in their manuscript fully available?

Reviewer #1: Yes

5. Is the manuscript presented in an intelligible fashion and written in standard English?

Reviewer #1: Yes

6. Review Comments to the Author

Reviewer #1: (No Response)

7. PLOS authors have the option to publish the peer review history of their article (what does this mean? ). If published, this will include your full peer review and any attached files.

**Do you want your identity to be public for this peer review?** For information about this choice, including consent withdrawal, please see our Privacy Policy .

Reviewer #1: No

---

## [Editor Report · Acceptance letter]

PONE-D-25-05271R1

PLOS One

Dear Dr. Lešinskis,

I'm pleased to inform you that your manuscript has been deemed suitable for publication in PLOS One. Congratulations! Your manuscript is now being handed over to our production team.

Kind regards,

on behalf of

Dr Ali Junaid Khan

Academic Editor

PLOS One